# Mitigating Tropical Forest Fragmentation with Natural and Semi-Artificial Canopy Bridges

**Diego Balbuena [1,2], Alfonso Alonso [1], Margot Panta [3], Alan Garcia [4] and Tremaine Gregory [1,\*]**

1   Center for Conservation and Sustainability, Smithsonian Conservation Biology Institute, National Zoological Park, Washington, DC 20013-7012, USA; Diego_Blb@live.com (D.B.); alonsoa@si.edu (A.A.)
2   Universidad Nacional Agraria La Molina, Av. La Molina s/n Apartado 12-056, Peru
3   Walsh Perú S.A. Ingenieros y Científicos Consultores, Alexander Fleming 187 Urbanización Higuereta, Surco, Lima 33, Peru; mpanta@walshp.com.pe
4   Repsol Exploración Perú, Sucursal del Perú, Víctor Andrés Belaunde 147, vía Principal 103, Oficina 202 San Isidro, Lima 27, Peru; alanmarlon.garcia@repsol.com
\*   Correspondence: gregoryt@si.edu

**Abstract:** Fragmentation caused by linear infrastructures is a threat to forest-dwelling wildlife globally. Loss of canopy connectivity is particularly problematic for highly arboreal species such as those of the Neotropics. We explored the use of both natural canopy bridges (NCBs) and a semi-artificial one over a natural gas pipeline right-of-way (RoW) in the Peruvian Amazon to provide more information on both a proven and a novel solution to the problem of fragmentation. We monitored seven NCBs over 14 months and found crossing rates higher than previously recorded (57.70 crossings/100 trap nights by 16 species). We also constructed a semi-artificial canopy bridge (SACB) out of a liana and found it to be used quickly (seven days after installation) and frequently (90.23 crossings/100 trap nights—nearly nightly) by five species (two procyonids, one didelphid, one primate, and one rodent). This information contributes to our knowledge of mitigation solutions for fragmentation. As linear infrastructure grows globally, more solutions must be developed and tested.

**Keywords:** linear infrastructure; natural gas pipeline; connectivity; camera trap; Peruvian Amazon

## 1. Introduction

Across the globe, human development is fragmenting and destroying ecosystems at an increasingly alarming rate [1]. Roads and other linear infrastructure elements are a particularly problematic threat, opening access for human settlements, logging, hunting, and mining (e.g., [2–7]). It is important that the impacts of these activities be evaluated so that they can be mitigated to the fullest possible extent [8,9]. As deforestation in the Amazon increases, in some cases reaching historic levels [6], the need for creative and scientifically tested mitigation methods grows.

To build a linear infrastructure such as a pipeline in a forested area, a right-of-way (RoW) must be opened to accommodate the laying of the pipe. An RoW is a linear clearing that provides access for the machinery that will build the pipeline, and, like a road, it divides the canopy. The resulting forest fragmentation has the potential to negatively affect some animals—particularly arboreal species that are unlikely to cross on the ground—because access to known feeding resources, sleeping sites, and potential mates, for example, may be lost and major consequences such as genetic isolation may ensue [10,11]. Canopy bridges (connections in the canopy composed of either artificial materials or branches) have proven to be an effective way to mitigate this form of forest fragmentation and reduce canopy connectivity loss or restore connectivity in forested areas impacted by a linear infrastructure (e.g., [10–16]).

Artificial canopy bridges (ACBs) have been implemented more commonly than natural ones to restore lost forest connectivity; however, they can be expensive [17]. Different types of bridges have been built for different animals, ecosystems, and types of infrastructure. They can be simple, consisting of wooden poles [18], single ropes, or bamboo [19–21], or more complicated with wooden support poles and cables [14] or complex rope configurations [12,13,22,23]. They can also be extremely expensive, using plastic coated steel cables and heavy machinery for transport or installation [17,24].

Trees on either side of the forest gap that connect by way of their branches form natural canopy bridges (NCBs). NCBs may have advantages over ACBs in reduced cost and habituation time due to animals' familiarity with the substrate and the possibility that the branches could be part of pre-existing paths. However, different project contexts likely call for different bridge solutions [10]. For example, NCB preservation may not be considered in pipeline construction, causing the need for a post-hoc solution. Alternatively, natural canopy connectivity by way of large trees with trunks sufficiently far apart to accommodate the RoW may not exist, or a project budget may have insufficient funding to allow for the construction of ACBs. Therefore, hybrid solutions that involve local materials familiar to animals and that involve reduced costs may provide a valuable solution in some contexts.

In this 14-month long study, we examined the utility of NCBs to mitigate tropical forest fragmentation caused by the construction of a natural gas pipeline. We also evaluated the use of a bridge made out of a naturally occurring liana strung "artificially" across the RoW clearing—what we are calling here a semi-artificial canopy bridge (SACB). Our objectives were the following: 1) verify the hypothesis that NCBs are used frequently and by a broad diversity of arboreal mammals to cross from one side of a fragmented forest to another and 2) test the use of an SACB made out of a naturally occurring liana by arboreal mammals.

## 2. Materials and Methods

### 2.1. Study Site

The study site is located in a natural gas concession (Block 57, Figure 1) managed by Repsol Exploración Peru (REP) in the province of La Convención, Cusco, in the Lower Urubamba Region (LUR) of the Peruvian Amazon (11°27′ S, 73°18′ W). The site is within the buffer zones of the Machiguenga Communal Reserve, Otishi National Park, and Ashaninka Communal Reserve. The RoW falls within the territories of three indigenous communities (Nuevo Mundo, Kitepampani, and Porotobango, with 783, 74, and 64 inhabitants, respectively, Figure 1), and hunting by community members is legal. The pipeline is located in a forested area 340–420 m in elevation, dominated by semi-dense forest (60–80% canopy cover) with patches of bamboo (*Guadua* spp.). The most abundant tree families in forested areas are Euphorbiaceae, Arecaceae, and Fabaceae [25]. The annual rainfall in the LUR is between 3000 and 3500 mm, while the average annual temperature is 27 °C and the average annual humidity is 80% [26]. The study took place along an 18.5 km stretch of a natural gas pipeline called Sagari. The pipeline RoW was constructed between March and October 2017 and was 16 m wide, on average.

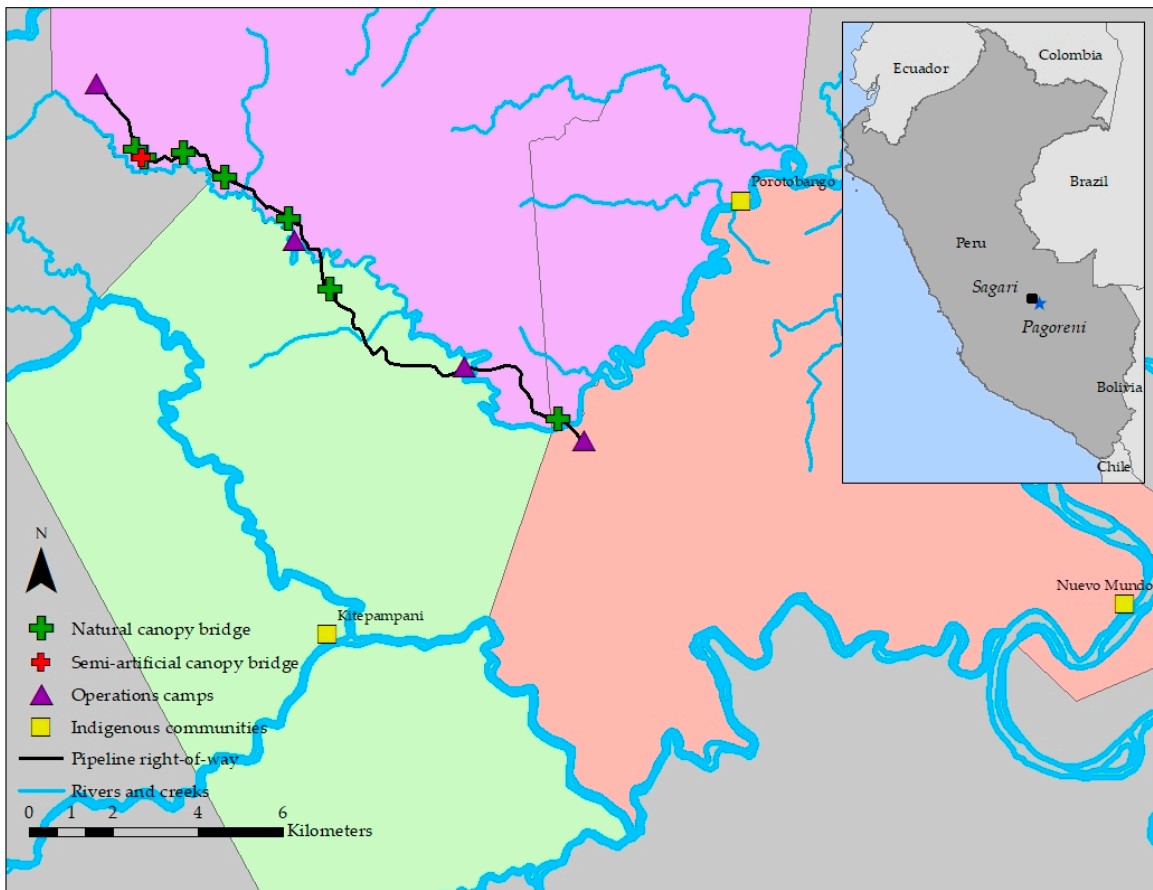

**Figure 1.** Map of the study area at Sagari, including the 18.5 km piece of pipeline right-of-way (RoW) (black line). The locations of both the natural canopy bridges (NCBs) (green crosses) and the semi-artificial canopy bridge (SACB) (red cross) are indicated, as are the locations of the operations camps (purple triangles) and nearest indigenous communities (yellow squares) and their respective territories (polygons). In the inset, the locations of Sagari (black square) and Pagoreni (blue star ~25 km to the southeast), another canopy bridge research location mentioned in the Discussion is labeled.

## 2.2. Bridge Selection

Identification of future potential NCB locations began in 2016 during the pipeline's design phase. The next year, before clearing the RoW, a team of biologists accompanied by members of REP's Operations and Health, Safety, and Environment branches, along with the pipeline construction contractors' engineers and topographers surveyed the RoW to verify the suitability and mark locations of potential candidate NCBs guided by bridge-resilience and engineering recommendations made by Gregory et al. [27]. Those recommendations included selecting bridge trees in good condition, with a diameter at breast height >50 cm, with few to no broken branches or trunk damage, and with a good connection to the surrounding forest. Engineering stipulations for bridge tree selection included a minimum of 8 m between bridge tree trunks and a minimum of 6 m in height of the connecting branches to allow heavy machinery to pass below safely, as well as avoiding bridges on slopes, curves, or hills [27]. Candidate bridge location selection was based solely on bridge-resilience criteria and engineering feasibility and not on behavioral factors. It was not possible to monitor arboreal mammals before impact to assess preexisting travel paths and territories. Even so, we imagine that doing such monitoring could be exceedingly challenging, potentially requiring radio collaring of multiple species and groups, methods far outside the scope of this study.

A total of 68 potential bridges were initially identified. Because of engineering constraints, of those, only seven bridges (Figures 1 and 2a) could be preserved during construction, and those were

monitored for the duration of the study. By the end of the bridge monitoring period in November 2018, two of the seven bridges had broken (one on June 20, 2018 and the other between May (when the camera batteries died) and November 2018, see Table S1, Supplementary Materials, for details). In addition, the two camera traps monitoring one of the bridges (NCB2) could not be recovered during camera retrieval because the local indigenous community denied access at the time of data collection.

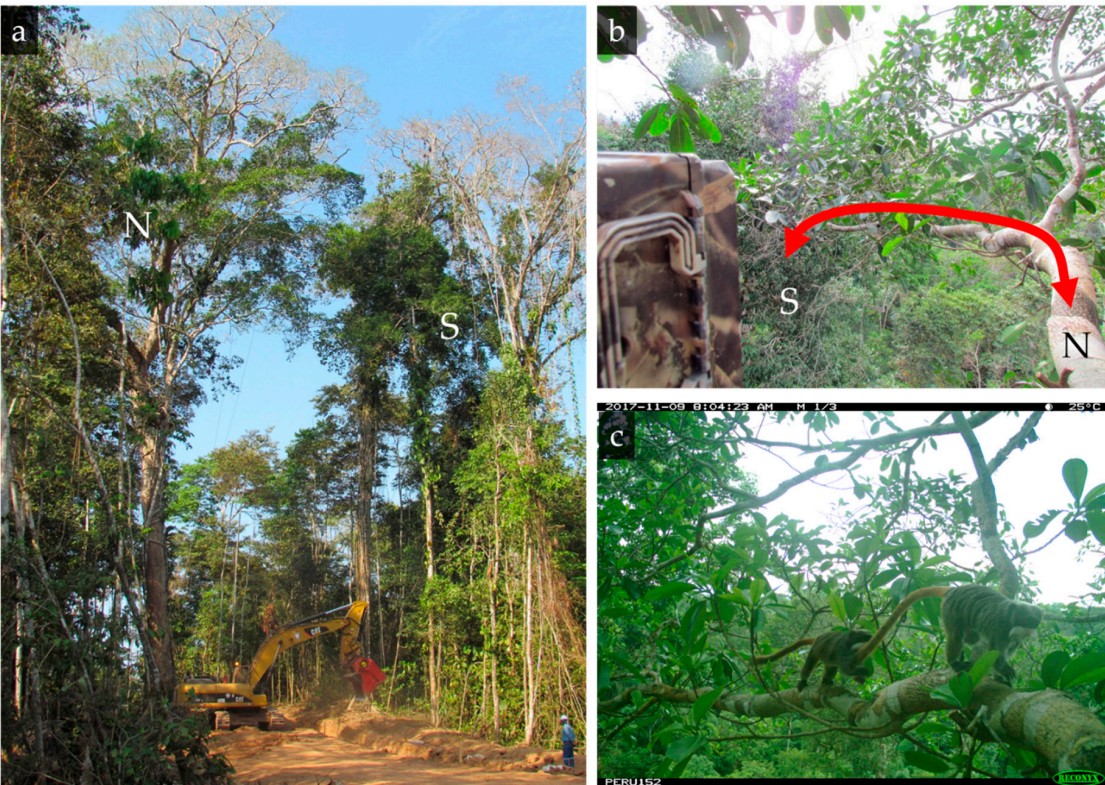

**Figure 2.** (**a**) One of the NCBs that was monitored with three camera traps for the three connections of branches between the tree on the north (N) and south (S) sides of the RoW, (**b**) the orientation of one of the three camera traps (see camera in the bottom left of the image) in the bridge, with the travel path it captured indicated by the red arrow, and (**c**) an example of a camera trap image of a crossing event by two emperor tamarin monkeys (*Saguinus imperator*).

The trunks of the pairs of trees that composed the NCBs were an average of 13.9 m apart (range of 11–18.8 m) and the average height of the lowest branches composing the bridges was 28.7 m (range of 21.5–32.5 m). The roots of most trees that composed the bridges experienced some damage during the opening of the RoW. The average distance between bridges was 2220 m (range of 220–8200 m).

### 2.3. Semi-Artificial Canopy Bridge

On 14 August 2017, after the RoW was cleared (but while the pipeline was still being laid), we found the optimal context in which to create an SACB across the RoW using a liana (location indicated in Figure 1), to evaluate whether animals would respond favorably to a bridge fabricated with a natural material. The liana had been growing naturally on the north side of the RoW when it was severed at its connection with the ground during clearing. We could not confirm identification of the liana species but suspect it to be a Bignoniaceae based on photos taken from the ground of the leaves. Using one-inch-wide black nylon webbing, we created a braided hitch over the last ~70 cm of the tip of the liana to hoist it across the clearing and secure it to a tree on the south side of the RoW (Figure 3a,b). Using a large slingshot, we ran a line of paracord tied to the webbing up over a branch on the south side, then three team members pulled the paracord to raise the liana while the fourth assisted from the

canopy. We then tied the paracord to a fallen tree trunk on the ground to secure it. The liana measured approximately 19.5 m in length, 6 cm in diameter, and 22.5 m high at its center point when suspended; the trunks were 25.5 m apart where it hung.

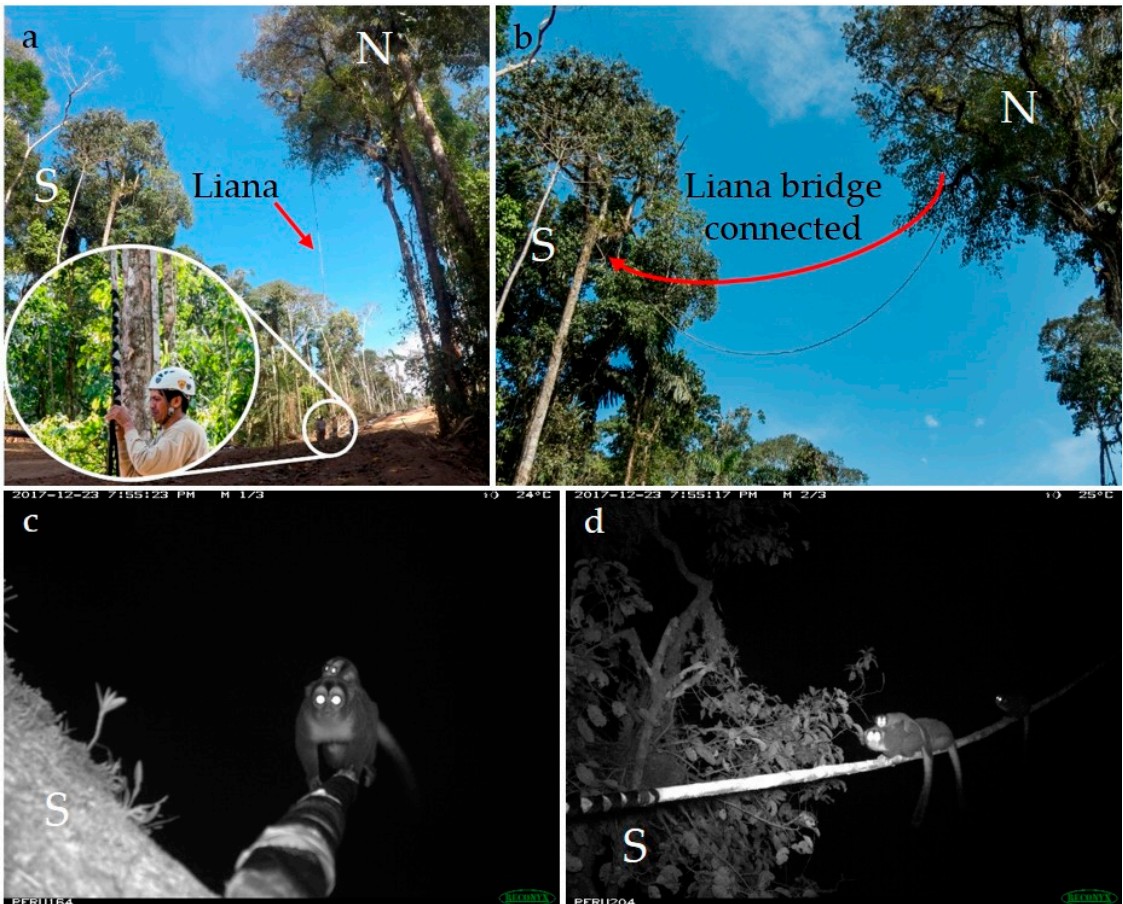

**Figure 3.** Installation of the SACB and resulting camera trap images of a crossing; N and S refer to the north and south sides of the RoW, respectively: (**a**) securing the braided nylon webbing hitch at the tip of the liana, (**b**) suspending the liana by pulling down the paracord line attached to the webbing (red arrow shows final location of the liana), (**c**) angle of head-on camera facing the liana, capturing a crossing by a family of night monkeys, (**d**) angle of the side-view camera, capturing the same crossing as (**c**).

### 2.4. Canopy Bridge Monitoring

Using camera placement methods described by Gregory et al. [28], we placed 12 Reconyx PC800 Hyperfire<sup>TM</sup> (Reconyx Inc., Holmen, WI, USA) cameras in the canopy in August 2017. The connectivity between the trees on either side of the RoW often involved more than one pair of branches. Therefore, we placed one camera at each connecting point (1–3 cameras per bridge, Table S1, Supplementary Materials) to capture all crossings (Figure 2b,c). We selected the connecting tree best for climbing safety (on either the north or south side of the RoW), and placed the cameras in that tree facing the different respective directions necessary to capture all crossings (Figure 2b,c).

In addition, we placed two cameras in the canopy on the south side (the more climbable side) of the RoW to monitor the liana bridge. In this case, unlike the natural bridges, in which one camera was used per connection, both cameras faced the single connection point but from two angles, ensuring "capture" of all crossings (Figure 3c,d).

We programmed all cameras to take three pictures per trigger, with no delay between triggers and a picture size of 3.1 MP. We used the Smithsonian's eMammal program, which divides sequences

using a one-minute rule, to evaluate the camera trap photos. After species identifications were made, they were verified during a second evaluation to ensure accuracy. Events were classified as "cross" or "unknown" depending upon whether it was possible to confirm if crossing occurred. We deduced that a crossing had occurred if the animal was clearly traveling in one direction (either from the crossing point or toward it) and did not double back.

The cameras were secured to the branches using the mount design described in Gregory et al. [28], with a double ball joint to allow for versatile positioning of the cameras. Each camera had an 8GB SD card, six AA lithium batteries, and a small bag of silica in the vacant second battery bay. After setup, we checked and maintained the cameras in November 2017 and March 2018 (battery, SD card, and silica changed) and removed them in November 2018.

## 3. Results

### 3.1. Canopy Bridge Monitoring

Over the 14 months and 4390 total trap days/nights ("trap nights" hereafter refer to a 24-hour cycle, days/nights in which a camera had stopped functioning or a bridge had broken were excluded from the count) of the study, we recorded a total of 16 mammal species crossing the seven natural bridges during 2533 crossing events (57.70 events/100 trap nights) (Table 1 and Figure S1, Supplementary Materials; 34 bird species and 4 reptile species were also recorded, see Table S2, Supplementary Materials). In addition, we recorded 1055 events (29.4% of all events recorded) in which crossing could not be verified (unknown in Table 1). There was a large difference in use between bridges, with the range of event rates being 14.9–279.5 crossing events/100 trap nights (mean rate = 80.9), and the range of the total number of mammal species using a bridge either day or night was 3–11 species (mean = 7.3 ± 2.9, Figure 4).

We did not evaluate the crossing rate relative to the distance between bridges to explore the question of potential increased use with further inter-bridge distance because the sample size was too low. For the same reason, we did not perform analyses comparing use of the two bridge types.

**Table 1.** Nocturnal and diurnal events of both crossing and unverifiable crossing (unknown) and rates (events/100 trap nights) in NCBs.

| Family | Species | | Events (rate/100 trap nights) | | | |
| | | | Cross | | Unknown | |
| | Scientific Name | Common Name | Nocturnal Events | Diurnal Events | Nocturnal Events | Diurnal Events |
|---|---|---|---|---|---|---|
| Aotidae | *Aotus nigriceps* | Black-headed night monkey | 660 (15.03) | | 264 (6.01) | |
| Atelidae | *Alouatta seniculus* | Red howler monkey | | 14 (0.32) | | 17 (0.39) |
| Cebidae | *Cebus albifrons* | White-fronted capuchin | | 1 (0.02) | | |
| Cebidae | *Saguinus imperator* | Emperor tamarin | | 50 (1.14) | | 22 (0.50) |
| Cebidae | *Sapajus macrocephalus* | Brown capuchin | 11 (0.25) | 195 (4.44) | | 7 (0.16) |
| Cyclopedidae | *Cyclopes didactylus* | Silky anteater | 2 (0.05) | | 3 (0.07) | |
| Didelphidae | *Caluromys lanatus* | Brown-eared woolly opossum | 409 (9.32) | | 54 (1.23) | |
| Didelphidae | *Didelphis marsupialis* | Common opossum | 6 (0.14) | | 7 (0.16) | |
| Didelphidae | *Marmosa* sp. | Mouse opossum species | 24 (0.55) | | 3 (0.07) | |
| Echimyidae | *Mesomys* sp. | Spiny tree rat species | 3 (0.07) | | 2 (0.05) | |
| Erethizontidae | *Coendou bicolor* | Bicolored-spined porcupine | 9 (0.21) | | | |
| Erethizontidae | *Coendou ichillus* | Streaked dwarf porcupine | 91 (2.07) | | 17 (0.39) | |
| Procyonidae | *Potos flavus* | Kinkajou | 706 (16.08) | | 194 (4.42) | |
| Procyonidae | *Bassaricyon alleni* | Eastern lowland olingo | 337 (7.68) | | 46 (1.05) | |
| Sciuridae | *Hadrosciurus spadiceus* | Southern Amazon red squirrel | 2 (0.05) | 5 (0.11) | | 1 (0.02) |
| Sciuridae | *Notosciurus pucheranii* | Andean squirrel | | 4 (0.09) | | |
| Unknown mammal | | | 4 (0.09) | | 410 (9.34) | 8 (0.18) |
| Total events (overall rates) | | | 2264 (51.57) | 269 (6.13) | 1000 (22.78) | 55 (1.25) |
| Total events (overall rates) | | | 2533 (57.70) | | 1055 (24.03) | |

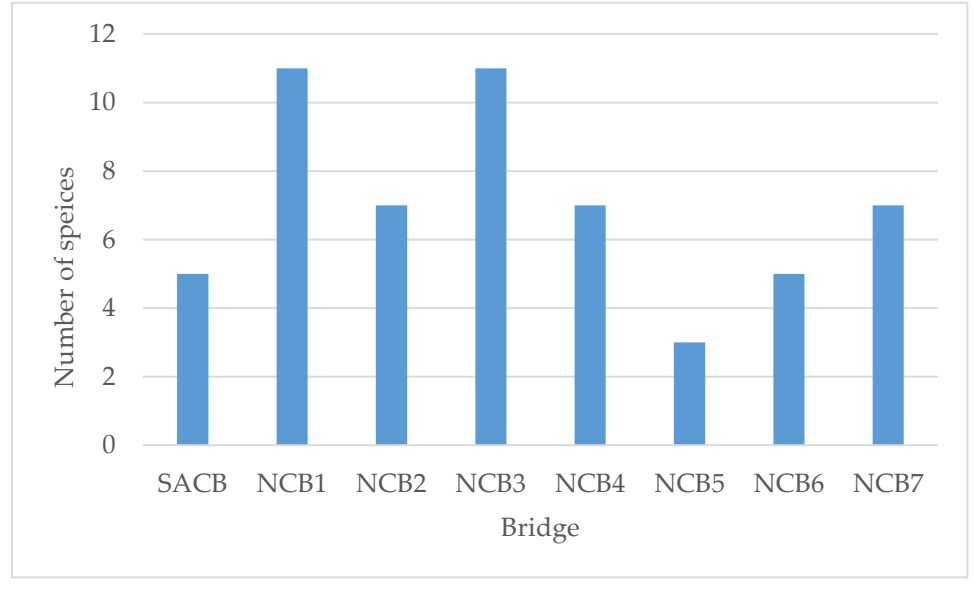

(a)

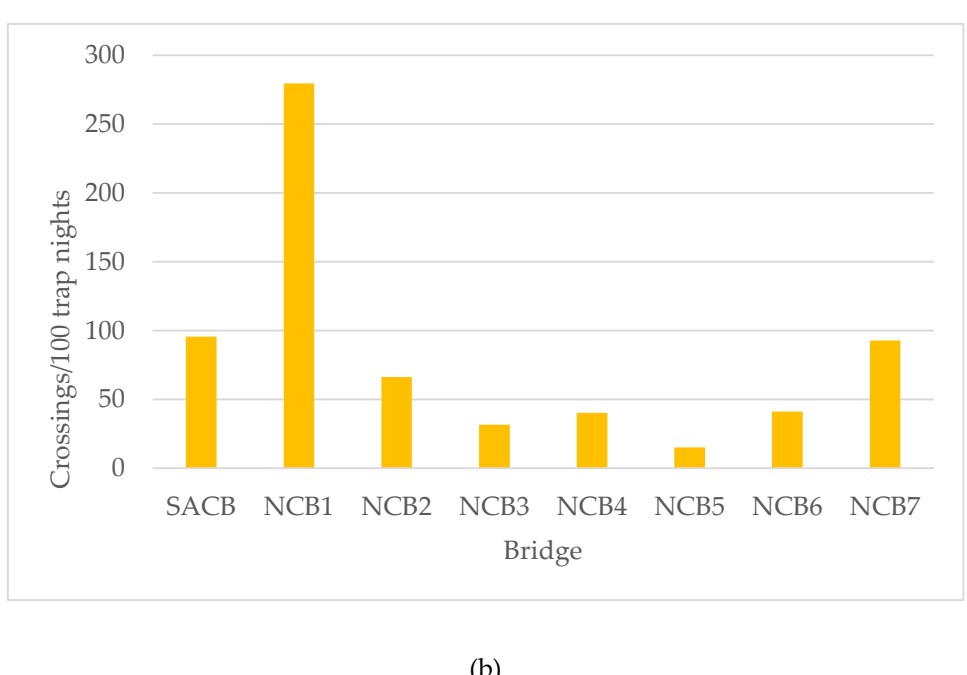

(b)

**Figure 4.** (**a**) The number of species and (**b**) the crossing rates in the SACB and the NCBs over the course of the study.

It is important to note that, in five cases, the memory cards filled before the camera was checked, in three cases, the batteries died before the camera was checked, and in one case, the camera malfunctioned from the date of installation (see Table S1, Supplementary Materials, for more detail). In one of these cases, the bridge eventually broke, but data were lost between card filling and bridge breaking, and in the other cases, data were lost from the connection being monitored by the full-card camera. In addition, data could not be collected from the two cameras in NCB2 for March–November 2018. Data loss due to camera malfunction is a danger in all camera trapping projects, but as noted by Gregory et al. [28] the probability of a card filling or the batteries dying in a tropical forest is higher in the canopy than on

the ground because of increased camera triggering by sun-warmed leaves that move in the wind and increased exposure to the elements.

The four most common species to use the bridges were the kinkajou (*Potos flavus*), black-headed night monkey (*Aotus nigriceps*), brown-eared woolly opossum (*Caluromys lanatus*), and eastern lowland olingo (*Bassaricyon alleni*), and the vast majority (88.5%) of crossings occurred at night (Table 1, Figure S1, Supplementary Materials). We also recorded the rare pygmy anteater (*Cyclopes didactylus*) using an NCB five times. Four events happened on the same early morning (20 September 2017, at 2:33 A.M., 2:43 A.M., 2:59 A.M., and 3:05 A.M.) and the fifth occurred four months later (25 March 2018, at 9:15 P.M.) (Figure S2, Supplementary Materials, in only two events could crossing be verified). This is the first record of the species crossing an NCB and the first record of the species for this study site [29–32], which is approximately 45 km outside of the current known range of the species [33].

### 3.2. Semi-Artificial Canopy Bridge

The SACB was used for the first time on August 21, 2017, just seven days after installation by an eastern lowland olingo (*Bassaricyon alleni*). In total, six mammal species explored the bridge and five used it to cross over 203 trap nights of two cameras for a total of 194 crossings (95.57 crossings/100 trap nights) (Table 2). The liana remained intact until March 6, 2018, when it broke when it became excessively dry and brittle.

**Table 2.** Events and rates (events/100 trap nights; 203 nights total) of mammal species recorded on the SACB, either crossing or exploring the bridge without a confirmed crossing (unknown). All events occurred at night except for one of the two events for *Sapajus macrocephalus*.

| Family | Species | | Events (rate/100 trap nights) | |
|---|---|---|---|---|
| | Scientific Name | Common Name | Cross | Unknown |
| Aotidae | *Aotus nigriceps* | Black-headed night monkey | 36 (17.73) | 6 (2.96) |
| Cebidae | *Sapajus macrocephalus* | Brown capuchin | | 2 (0.99) |
| Didelphidae | *Caluromys lanatus* | Brown-eared woolly opossum | 2 (0.99) | |
| Echimyidae | *Mesomys* sp. | Spiny tree rat species | 29 (14.29) | 21 (10.34) |
| Procyonidae | *Bassaricyon alleni* | Eastern lowland olingo | 95 (46.80) | 14 (6.90) |
| Procyonidae | *Potos flavus* | Kinkajou | 28 (13.79) | 4 (1.97) |
| Unknown mammal | | | 4 (1.97) | 9 (4.43) |
| Total events (overall rates) | | | 194 (95.57) | 56 (27.59) |

## 4. Discussion

Different types of linear infrastructure crossing structures—including NCBs, ACBs, glider poles, overpasses, and underpasses—have been developed on every continent but Antarctica to restore or maintain landscape connectivity either for specific species, solely for arboreal animals, or for entire terrestrial and/or arboreal communities in different landscapes (e.g., [10,19,22,34–36]). Some solutions are more effective than others [37], but as the global infrastructure "tsunami" escalates [7], so must our efforts to reduce its fragmentation effects.

Besides this study, others have confirmed the use of both natural [10,15,16,20] and artificial [12–14,17–19,22,24,34,36,38,39] canopy bridges. This study highlights their use and value connecting a clearing that was up to 18.8 m wide. We found overall use rates of the bridges to be over once every two nights, but as in other studies [22,36], we found a broad difference in use rates between crossing structures, with some crossed multiple times a night and others crossed closer to once a week (Figure 4). We also found slightly higher use rates than at a similar site in the LUR [Gregory et al. [10] found 44.47 crossings per 100 trap nights in Pagoreni], and while the most frequent use species were similar to Pagoreni, two were detected using canopy bridges for the first time (*Alouatta seniculus* and *Cyclopes didactylus*).

Perhaps the most salient conclusion of this study is the finding that a canopy bridge made out of a single liana was used very promptly upon installation and relatively frequently (nearly once

a night). While one other study found habituation to be nearly immediate [18], other studies have found habituation by arboreal mammals to artificial bridges to take months for some species [23,39,40]. In the present study, not only was the SACB used quickly, but it was composed of a single, narrow line bridging a considerable gap with no surrounding vegetation. We hypothesize that attraction to the bridge is attributable to the fact that it is a familiar substrate, unlike a rope or cable bridge. In fact, in an experiment with primates (*Alouatta paliatta* and *Ateles geoffroyi*) in a sanctuary in Costa Rica, Narváez Rivera and Lindshield [21] found animals preferred a natural substrate—bamboo—over others (rope, etc.) when crossing a small clearing. We fully recognize, however, that the liana broke after approximately seven months and therefore on its own may not be a good option. However, where an RoW is narrower, lianas may have a better chance of survival or other species of lianas may provide better options. Furthermore, hybrid bridges, made out of a liana intertwined with, for example, steel cable, may be a valuable design to test.

It is important to note that forest types around the RoW and the quality or degree of connectivity of the canopy likely affected how many bridges could be left in Sagari. At this study site, 53% of the forest was dominated by semi-dense (>60% canopy cover) forest with bamboo patches and sparse forest with bamboo [29]. Because NCBs generally must be composed of large trees that can span the gap, there were not very many opportunities for them to be left intact. Therefore, when NCBs are not a viable possibility, it is valuable to have ACB options. For this reason, studies evaluating the use of different materials and designs are needed. The SACB we installed in the present study not only had the benefit of being a familiar substrate for animals, but it was also a relatively inexpensive option because it required very little equipment (paracord line, webbing, climbing equipment, and a camera trap) and installation cost (approximately five hours by four people). However, as we saw here, a natural material may not withstand the strain of being suspended over a wide clearing for a long time, making materials such as cable, rope, webbing, among others, preferable materials. On the other hand, the expense of a cable/rope bridge is likely to be much higher (estimated for Sagari to be 40% higher than an NCB), not only because of the building of the bridge itself, but also because of the cost of transporting the bridge materials to a remote location (i.e., a helicopter flight may be needed depending on accessibility and weight of the bridge) and installing it (i.e., special equipment or climbing expertise may be needed if the bridge is extremely heavy or complex to install) [M. Thurber, pers. comm., 17].

Another important consideration in choosing between bridge options in fragmentation mitigation projects with linear infrastructure is understanding the life cycle of a project. Depending on the projected period during which a clearing will be present, different options may be suitable. For example, RoWs are often reforested after a pipeline is placed and construction is complete, as was the case for both Sagari and Pagoreni. This allows for connectivity of the forest on either side of the RoW to eventually be restored, reducing the need over time for the NCBs. Being a permanent structure, a road would require a long-term solution. Temporary fragmentation caused by an RoW presents an opportunity for testing how well different bridge types can reduce long-term fragmentation effects caused by roads.

We recommend that future studies continue to focus on ACB designs and substrates, including potential ways to reinforce ACBs composed of natural substrates (e.g., intertwining with cable). We also suggest that NCB research seek to address the question of an appropriate distance between bridges. Gregory et al. [10] suggest bridges be <300 m apart along an RoW, but an experiment addressing this question would be a valuable contribution. Bissonette and Adair [41] developed metrics for determining viable inter-crossing distances for terrestrial North American mammals; similar methods could be applied to evaluations of arboreal species' connectivity needs. Finally, while understanding travel paths and the locations of territories of groups of arboreal mammals within the area of influence of the RoW was outside the scope of this study, ACB and NCB effectiveness could be further maximized by studies to address these points. Perhaps forms of remote sensing technology, such as drone monitoring using infrared sensors [42], will allow these points to be addressed one day.

**Supplementary Materials:** The following are available online at http://www.mdpi.com/1424-2818/11/4/66/s1, Figure S1. The four species that used the NCBs the most: (**a**) *Potos flavus*, (**b**) a group of three *Aotus nigriceps*, (**c**) *Caluromys lanatus*, and (**d**) *Bassaricyon alleni* with offspring, Figure S2. Pygmy anteater (*Cyclopes didactylus*) crossing an NCB, Table S1. Information on canopy bridges and their maintenance over the course of the 14-month study, Table S2. Bird and reptile species detected in camera trap images in the NCBs and the SACB, with the number of events and the trapping rate indicated.

**Author Contributions:** Conceptualization, T.G., D.B., M.P., A.G., and A.A.; methodology, T.G., D.B., and M.P.; formal analysis, D.B. and T.G.; investigation, D.B. and T.G.; resources, M.P., A.A., and A.G.; data curation, D.B. and T.G.; writing—original draft preparation, T.G., D.B., and M.P.; writing—review and editing, A.A. and A.G.; project administration, M.P., A.G., T.G., and A.A.; funding acquisition, M.P., A.G., T.G., and A.A.

**Funding:** This research was funded by Repsol Exploración Perú.

**Acknowledgments:** We thank researchers from GEMA Peru, led by Narda Casaverde, for their contribution to bridge identification. We thank the team from eMammal led by Jen Zhao for support with data analysis. We also thank Walsh Perú S.A.'s environmental research management team including Nadia Sanchez and Wendy Calderón, and we thank Piter Ccoicca, Geiser Gutierrez, Ernesto Vargas Ríos, Rody Peña Vasquez, Denis Pacaya Perez, and Aldo Campos Ríos for their invaluable field support and Tatiana Pacheco for administrative support. Finally, we thank Jessica Deichmann and three anonymous reviewers for their valuable comments on previous drafts of this manuscript, and Reynaldo Linares for botanical identification assistance. This is contribution #58 of the Peru Biodiversity Program led by SCBI.

**Conflicts of Interest:** The authors declare no conflict of interest. REP funded this study; however, all aspects of the design, methods, and data interpretation were conducted by Smithsonian and Walsh staff. Company personnel were involved in managing logistics and personal safety during fieldwork. An REP staff member (A.G.) also contributed to the development of this manuscript, but he confirms no conflict of interest.

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
