# Peer review of "Mitigating Tropical Forest Fragmentation with Natural and Semi-Artificial Canopy Bridges"

_diversity, doi:10.3390/d11040066_

Round 1

Reviewer 1 Report

This manuscript provides details of a camera trap study exploring the use of canopy bridges between areas of tropical forest recently fragmented by clearances for a potential gas pipeline. The study observes crossings by mammal species using natural bridges, deliberately left in place by the construction company, and a single man-made bridge built using a liana cut during the clearance described. This is an interesting and potentially valuable study but it would greatly benefit from greater attention throughout the manuscript. This includes broadening the Introduction and Discussion to explain the context of this study within the issue of forest fragmentation in the face of growing development pressure. In addition, further details in the Methods and a more comprehensive presentation (and analysis!) of the Results are essential.

"Abstract L22: Repetition of ‘by five species’. Include details of which taxonomic groups were recorded? Keywords Duplication of ‘canopy bridge’ from title. Suggest ‘connectivity’ as possible alternative. Introduction A general introduction to the growing pressure of development projects in tropical forests would be beneficial here. There are several recent papers that would be useful starting points here e.g. Alamgir et al. 2019 Scientific Reports 9; Laurance 2018 TREE 33. L29-31: Is this a general requirement in all countries, or is this ‘right-of-way’ specific to Peru? This terms seems to contrast to another sense of right-of-way, which would refer to access by the public, as opposed to the physical clearance of a linear strip of forest that is implied here. L32: Which sort of resources? L33: These definitions are unclear. The artificial bridge you use in this study is made of natural materials (lianas), suggesting that artificial bridges are defined by whether they occur naturally or are man-made, rather than the particular materials used to construct them. This contrasts with the implication here that it is the materials that distinguish between artificial and natural canopy bridges. Alternatively, and even more confusingly, the way that this sentence is currently written may imply that natural bridges can be made of artificial materials! L34-36: If natural canopy bridges are those that occur naturally (rather than those constructed by man but using natural materials) then existing natural bridges cannot ‘restore’ connectivity but only reduce connectivity loss. It would only be new canopy bridges constructed after fragmentation (using natural or artificial materials) that could potentially restore a degree of connectivity. Unless implying growth of branches or lianas across the new gap? L39-42: This sentence is rather long and could usefully be split into two, with the first focussing on simple vs complex structures, and the second focussing on cheap vs expensive. L46: What is meant by ‘different project contexts?’ L49: tropical forest fragmentation L54: Use of the word ‘project’ is vague and possibly not relevant here. It could easily refer to the original exploration activity i.e. installing the pipeline. It would therefore be useful to specify that you mean costs that are relevant for those wishing to install canopy bridges to mitigate connectivity loss following fragmentation. Methods L61: Do you mean Arecaceae? It would also be useful to include a general description of the forest type/category before giving examples of the most abundant families. L64: How far is this from other forms of fragmentation or disturbance? I.e. is this forest otherwise intact? Perhaps a map would be useful, at least as a supplementary figure. L77: Were all potential 68 bridges allowed to persist by the construction company? L79: Presumably, these were two of the seven monitored bridges? Better to state this clearly and avoid ambiguity – otherwise it could be read that two of the 68 bridges broke. Presumably you know when these bridges broke during that monitoring period? L84-85: This is important to include but perhaps here is not the most appropriate place. Better within a ‘data analysis’ section? L87: Earlier (L65) you state that the RoW construction took place between March and October 2017. August 2017 would therefore be during the construction phase, not after, as stated here. L87: Who is the ‘research team’? Presumably you, the authors of this manuscript? If so, suggest just using ‘we’. L90: Presumably this means 1 inch but does it mean the mesh size or the rope diameter? Check the format for writing units and be more specific about what the measurement refers to. L92-93: Would it be relevant to give the ID of the trees on either side? L94: Delete typo ‘It.’ L95: Is this the trunk of a live tree or the stump of a dead tree? L95: Do you know the species ID of the liana? L105: See previous suggestion to replace ‘research team’ with ‘we’. L106-107: Specify that this refers to the ‘seven natural canopy bridges’ L108: Why is it important that the cameras for the artificial bridge were on the south side? If this information is relevant, then which side were the cameras for the natural bridges? L109-110: But just before, you state that you used 1-3 cameras per bridge, not only 1 per bridge? L115-116: So, did you also have a category (e.g. ‘unknown’) for where it was not possible to identify if a crossing took place or not? L121: Were all cameras still functioning correctly when checked? Where there any failures (e.g. dead batteries, humidity-induced failure of cameras, full SD cards) that would mean missed monitoring periods? Results There does not appear to be any attempt to analyse the data collected in any statistical way. L124: What was the total number of trap-hours? Why the focus on nights? Presumably you were interested in diurnal as well as nocturnal crossings? L124: ‘16 animal species’ – or perhaps ’16 mammal species’, to be more specific. I.e. you do not include other animals e.g. reptiles, birds, insects etc. L126: Were the species also identified for these non-confirmed crossings? Were there any differences between confirmed and non-confirmed crossings? I.e. were there species that were recorded at the bridge but not confirmed to cross? For how many (what proportion of) records could the mammal species not be identified? L128: Again, why trap-night as the unit? What about the daytime? Does this term refer to the 12 hour nocturnal period or the full 24 hour day/night period? L128-129: 3-11 species per day/night or in total at that bridge? L137: But Table 1 only shows two events for this species? It seems a jump to suddenly start talking about results for this particular species. How about some more mention of the more common species first. These are presented in ‘Figure 2’ but not mentioned in the text. L140: Ok, so Table 1 only includes ‘confirmed crossings’ then? This needs to be made clearer. L145: You call this a liana bridge here but I suggest that Artificial Canopy Bridge would be more consistent with your previous description. L147: Suggest using common name, followed by scientific name. L154: If this section is to be included then it needs to be much more detailed. There is no breakdown of different costs and no financial figures presented at all. The proportional cost given is completely meaningless without knowing the total cost of the Sagari Project, which is undescribed and undefined. L159: The meaning of this sentence is not at all clear. Discussion It would be useful to widen the discussion to consider techniques employed in other regions of the world, including other habitats, to improve connectivity in fragmented landscapes e.g. corridors, tunnels, fence gaps etc. L178: This could have been tested (at least minimally) by also presenting an alternative ACB made with man-made materials. It is important to recognise that the natural material of a dead liana had a limited lifetime. So even, if having a quicker uptake than a bridge made of artificial materials, what is the long-term potential for an artificial bridge made of natural materials in any practical sense? (I see this point is recognised to some degree in L192 but the authors still maintain that the liana bridge is a viable option considering its lower (although unquantified) cost. The cost of constantly replacing a liana bridge would have to be considered here to give this comparison some more validity. L180: Which primates? L200-209: This situation is important to discuss in general but of less relevance to the more theoretical questions about connectivity. The temporary fragmentation caused by the RoW could perhaps be better seen as an opportunity to test how bridges could reduce the effects of fragmentation from road construction. Figures Figure 1: Very nice photos! Also useful to appreciate the methodology. Figure 2/3: These are nice photos too but may be more appropriate as supplementary figures. It may be more useful to instead include figures that present the main findings of the study. Tables Table 1: Need to specify in the table title that this only includes results from the Natural Canopy Bridges. Where is Pagoreni? If comparisons to this site comprise an important aspect of this study, then the details should also be provided in the Methods. Perhaps this site (if nearby) could be included in the suggested map figure. What was the sampling effort (number of bridges, number of camera traps, length of study period etc.) at this site? Do the rates represent events or numbers of individuals? I.e. did the group of night monkeys score as 1 event or did each individual score separately?  Suggest a more meaningful order for the species e.g. arranged taxonomically. Do the species names need associated common names and/or authorities? Table 2: This table for ACB presents data on crossing/not-crossing that seems to be missing for the NCBs. However, in the opposite manner, this table does not distinguish between diurnal and nocturnal crossings. Suggest greater attention to present comprehensive and consistent results. Supplementary Materials These were not accessible to the reviewer. Funding If funded by the construction company involved, does this not represent a potential conflict of interest?"

Author Response

AUTHOR RESPONSES IN CAPS

REVIEWER 1

               Yes         Can be improved              Must be improved           Not applicable

Does the introduction provide sufficient background and include all relevant references?

                                                                                          ( )           ( )           (x)           ( )

Is the research design appropriate?                         ( )           ( )           (x)           ( )

Are the methods adequately described?                ( )           ( )           (x)           ( )

Are the results clearly presented?                            ( )           ( )           (x)           ( )

Are the conclusions supported by the results?     ( )           ( )           (x)           ( )

Comments and Suggestions for Authors

This manuscript provides details of a camera trap study exploring the use of canopy bridges between areas of tropical forest recently fragmented by clearances for a potential gas pipeline. The study observes crossings by mammal species using natural bridges, deliberately left in place by the construction company, and a single man-made bridge built using a liana cut during the clearance described. This is an interesting and potentially valuable study but it would greatly benefit from greater attention throughout the manuscript. This includes broadening the Introduction and Discussion to explain the context of this study within the issue of forest fragmentation in the face of growing development pressure. In addition, further details in the Methods and a more comprehensive presentation (and analysis!) of the Results are essential.

THANK YOU FOR THIS VALUABLE FEEDBACK. WE HAVE MADE CHANGES AND ADDITIONS THROUGHOUT THE MANUSCRIPT ACCORDING TO YOUR FEEDBACK AND THAT OF THE OTHER REVIEWERS. PLEASE KEEP IN MIND THAT THIS IS A BRIEF COMMUNICATION, SO WE DID NOT ADD EXTENSIVE DETAIL. ALSO, PLEASE SEE OUR COMMENTS ON THE ANALYSIS BELOW. WE DO NOT FEEL THESE DATA LEND THEMSELVES TO STATISTICAL ANALYSES.

"Abstract L22: Repetition of ‘by five species’.

FIXED

Include details of which taxonomic groups were recorded?

FIXED, PLEASE INDICATE IF SPECIES NAME IS PREFERRED OVER FAMILY

Keywords Duplication of ‘canopy bridge’ from title. Suggest ‘connectivity’ as possible alternative.

FIXED

Introduction A general introduction to the growing pressure of development projects in tropical forests would be beneficial here. There are several recent papers that would be useful starting points here e.g. Alamgir et al. 2019 Scientific Reports 9; Laurance 2018 TREE 33.

WE ADDED A PARAGRAPH TO THE INTRODUCTION (THE FIRST ONE) TO ADDRESS THIS CONCERN.

L29-31: Is this a general requirement in all countries, or is this ‘right-of-way’ specific to Peru? This terms seems to contrast to another sense of right-of-way, which would refer to access by the public, as opposed to the physical clearance of a linear strip of forest that is implied here.

THE RESEARCH SITE IS LOCATED IN TROPICAL FOREST, AND HEAVY MACHINERY (BULLDOZERS, SIDE-BOOMS, DUMP TRUCKS, ETC) MUST BE BROUGHT IN TO LAY THE LONG STRINGS OF 16” STEEL PIPE. THE RIGHT-OF-WAY (ROW) IS NECESSARY FOR THE HEAVY MACHINERY TO PASS AND LAY THE PIPE IN THE TRENCH IN WHICH IT IS BURIED. IT WOULD NOT BE POSSIBLE TO LAY THE PIPE WITHOUT THE ROW. THE WIDTH OF THE ROW IS STIPULATED IN THE ENVIRONMENTAL IMPACT ASSESSMENT (EIA), WHICH IS PRODUCED BY THE COMPANY RESPONSIBLE FOR BUILDING THE PIPELINE. THE EIA IS REVIEWED AND APPROVED BY THE GOVERNMENT. RATHER THAN THE DEFINITION OF “RIGHT-OF-WAY” YOU PROVIDE ABOVE, IN THIS CONTEXT THE TERM REFERS TO THE CLEARING (ESSENTIALLY A TEMPORARY ACCESS ROAD) PRODUCED TO LAY THE PIPE. PLEASE SEE THIS REFERENCE WHERE THE TERM IS DEFINED (AND BOTH MEANINGS OF ROW ARE USED): http://www.sunocologistics.com/Public-Awareness-Safety/Right-Of-Way/What-is-a-Right-of-Way/121/

L32: Which sort of resources?

DETAIL ADDED IN THE TEXT. WE ARE REFERRING TO MULTIPLE TYPES OF RESOURCES BOTH FOR FEEDING, FOR SLEEPING, AND FOR MATING, AMONG OTHERS.

L33: These definitions are unclear. The artificial bridge you use in this study is made of natural materials (lianas), suggesting that artificial bridges are defined by whether they occur naturally or are man-made, rather than the particular materials used to construct them. This contrasts with the implication here that it is the materials that distinguish between artificial and natural canopy bridges. Alternatively, and even more confusingly, the way that this sentence is currently written may imply that natural bridges can be made of artificial materials!

THANK YOU FOR THIS OBSERVATION. WE HAVE FIXED THE TEXT THROUGHOUT THE PAPER AND NOW DEFINE THE LIANA BRIDGE AS A SEMI-ARTIFICIAL CANOPY BRIDGE. WE HOPE THE TEXT IS NOW CLEARER.

L3436: If natural canopy bridges are those that occur naturally (rather than those constructed by man but using natural materials) then existing natural bridges cannot ‘restore’ connectivity but only reduce connectivity loss. It would only be new canopy bridges constructed after fragmentation (using natural or artificial materials) that could potentially restore a degree of connectivity. Unless implying growth of branches or lianas across the new gap?

THANK YOU FOR NOTING THIS. WE HAVE CHANGED THE TEXT.

L39-42: This sentence is rather long and could usefully be split into two, with the first focussing on simple vs complex structures, and the second focussing on cheap vs expensive.

SENTENCE SPLIT AS SUGGESTED

L46: What is meant by ‘different project contexts?’

WE PROVIDE MORE DETAIL ON THIS POINT IN THE TEXT

L49: tropical forest fragmentation

CHANGED

L54: Use of the word ‘project’ is vague and possibly not relevant here. It could easily refer to the original exploration activity i.e. installing the pipeline. It would therefore be useful to specify that you mean costs that are relevant for those wishing to install canopy bridges to mitigate connectivity loss following fragmentation.

CHANGED

Methods

L61: Do you mean Arecaceae? It would also be useful to include a general description of the forest type/category before giving examples of the most abundant families.

FIXED. THANK YOU!

L64: How far is this from other forms of fragmentation or disturbance? I.e. is this forest otherwise intact? Perhaps a map would be useful, at least as a supplementary figure.

WE ADDED A MAP. THIS SITE IS IN CONTINUOUS FOREST, BUT WITHIN INDIGENOUS TERRITORIES.

L77: Were all potential 68 bridges allowed to persist by the construction company?

ONLY THE SEVEN STUDIED WERE SUCCESSFULLY PRESERVED. TEXT ADDED TO CLARIFY.

L79: Presumably, these were two of the seven monitored bridges? Better to state this clearly and avoid ambiguity – otherwise it could be read that two of the 68 bridges broke. Presumably you know when these bridges broke during that monitoring period?

WE FIXED THE TEXT HERE TO ADD CLARITY. WE CANNOT CONFIRM THE DATE OF BRIDGE BREAKAGE IN ONE CASE BECAUSE THE CAMERA BATTERIES RAN OUT BEFORE BREAKAGE

L84-85: This is important to include but perhaps here is not the most appropriate place. Better within a ‘data analysis’ section?

MOVED. THANK YOU.

L87: Earlier (L65) you state that the RoW construction took place between March and October 2017. August 2017 would therefore be during the construction phase, not after, as stated here.

THE CONSTRUCTION OF THE PIPELINE INVOLVES MANY STEPS. THE INTIAL STEPS INVOLVE OPENING AND CLEARING OF THE ROW. ONCE IT IS OPENED, THE DIGGING OF THE TRENCH FOR THE PIPE AND A LONG SERIES OF OTHER STEPS BEGIN. IN AUGUST, THE ROW OPENING WAS COMPLETE, BUT THERE WERE STILL MANY STEPS TO GO. CLARIFICATION ADDED.

L87: Who is the ‘research team’? Presumably you, the authors of this manuscript? If so, suggest just using ‘we’.

CHANGED

L90: Presumably this means 1 inch but does it mean the mesh size or the rope diameter? Check the format for writing units and be more specific about what the measurement refers to.

BY WEBBING, WE ARE REFERRING TO A STRAP-LIKE PRODUCT, NOT A WEBBED OR MESH-LIKE PRODUCT. SOMETHING LIKE THIS: https://en.wikipedia.org/wiki/Webbing. IT IS SOLD IN ONE-INCH WIDTHS. FOR THIS REASON WE USE INCHES RATHER THEN CENTIMETERS.

L92-93: Would it be relevant to give the ID of the trees on either side?

WE ADDED THE LETTERS “N” AND “S” TO THE FIGURE TO CLARIFY WHICH TREE WAS ON THE NORTH VS THE SOUTH SIDE OF THE ROW.

L94: Delete typo ‘It.’

DONE

L95: Is this the trunk of a live tree or the stump of a dead tree?

IT WAS A FALLEN TREE. BEING HORIZONALLY ORIENTED, IT WAS EASY TO TIE THE PARACORD TO.

L95: Do you know the species ID of the liana?

UNFORTUNATELY, NO. WE WERE NOT ABLE TO USE A PHOTO OF THE LEAVES FOR IDENTIFICATION BECAUSE LIANA DIVERSITY IS SO BROAD AND SOMEWHAT UNKNOWN IN THE AREA AND DISTINGUISHING WHICH ARE THE CORRECT LEAVES IN A PHOTO AMIDST THE LEAVES OF ALL OF THE EPIPHYTES WAS INHIBITINGLY CHALLENGING. WE SUSPECT IT TO BE A BIGNON AND INDICATED THIS IN THE TEXT.

L105: See previous suggestion to replace ‘research team’ with ‘we’.

FIXED

L106-107: Specify that this refers to the ‘seven natural canopy bridges’

“NATURAL” INSERTED

L108: Why is it important that the cameras for the artificial bridge were on the south side? If this information is relevant, then which side were the cameras for the natural bridges?

THE MAIN REASON WE PLACED THEM ON THE SOUTH SIDE WAS FOR CLIMBER SAFETY (SAFETY WAS THE PRIORITY FOR PLACEMENT OF CAMERAS IN EVERY BRIDGE), BUT WE INDICATED THIS IN THE TEXT TO MAKE THE FIGURE MORE UNDERSTANDABLE. WE HAVE ADDED “THE MORE CLIMBABLE SIDE” FOR CLARIFICATION.

L109-110: But just before, you state that you used 1-3 cameras per bridge, not only 1 per bridge?

WE PLACED ONE CAMERA PER CONNECTION, SO BRIDGES WITH MULTIPLE CONNECTIONS HAD MULTIPLE CAMERAS. WE ADDED CLARIFICATION IN THE TEXT.

L115-116: So, did you also have a category (e.g. ‘unknown’) for where it was not possible to identify if a crossing took place or not?

WE ARE USING UNKNOWN AS SUGGESTED.

L121: Were all cameras still functioning correctly when checked? Where there any failures (e.g. dead batteries, humidity-induced failure of cameras, full SD cards) that would mean missed monitoring periods?

IN TWO CASES THE MEMORY CARD FILLED BEFORE THE CAMERA WAS CHECKED. WE ADDED A PARAGRAPH WITH THIS INFORMATION TO THE RESULTS.

Results There does not appear to be any attempt to analyse the data collected in any statistical way.

WE DO NOT THINK THE DATA LEND THEMSELVES TO STATISTICAL ANALYSES DUE TO SMALL SAMPLE SIZES AND AN ABUNDANCE OF UNMEASURABLE VARIABLES; FOR EXAMPLE, THERE WERE ONLY SEVEN BRIDGES AND CONFOUNDING VARIABLES IMPOSSIBLE TO MEASURE INCLUDE WHETHER THE BRIDGE FELL INSIDE OF A PREXISTING TERRITORY OF A GROUP/SPECIES, WHETHER THE CONNETION TO THE REST OF THE FOREST WAS FAVORABLE TO CROSSING BY ALL SPECIES, ETC.

L124: What was the total number of trap-hours? Why the focus on nights? Presumably you were interested in diurnal as well as nocturnal crossings?

TRAP NIGHTS IS A STANDARD FORM OF MEASURMENT IN CAMERA TRAPPING STUDIES (SEE ROVERO AND ZIMMERMAN 2016 Camera Trapping for Wildlife Research). IT REFERS TO A 24 HOUR PERIOD, SO IT COULD BE CALLED A TRAP DAY OR A TRAP DAY/NIGHT, BUT WE USED TRAP NIGHT DUE TO ITS COMMON USEAGE. WE ADDED CLARIFICATION IN THE TEXT.

L124: ‘16 animal species’ – or perhaps ’16 mammal species’, to be more specific. I.e. you do not include other animals e.g. reptiles, birds, insects etc.

WE ADDED “MAMMAL” SPECIES FOR CLARIFICATION. (INDEED WE HAD EVENTS OF BIRDS AND REPTILES NOT PRESENTED HERE)

L126: Were the species also identified for these non-confirmed crossings? Were there any differences between confirmed and non-confirmed crossings? I.e. were there species that were recorded at the bridge but not confirmed to cross? For how many (what proportion of) records could the mammal species not be identified?

WE ADDED THIS INFORMATION TO TABLE 1.

L128: Again, why trap-night as the unit? What about the daytime? Does this term refer to the 12 hour nocturnal period or the full 24 hour day/night period?

AGAIN, THIS TERM REFERS TO A 24 HOUR PERIOD. SEE RESPONSE TO COMMENT ON L124

L128-129: 3-11 species per day/night or in total at that bridge?

SEE THE COMMENT ABOVE. THIS REFERS TO THE TOTAL NUMBER OF MAMMAL SPECIES THAT USED THE BRIDGE EITHER DAY OR NIGHT. WE ADDED CLARIFICATION.

L137: But Table 1 only shows two events for this species? It seems a jump to suddenly start talking about results for this particular species. How about some more mention of the more common species first. These are presented in ‘Figure 2’ but not mentioned in the text.

WE ADDED MORE INFORMATION HERE ABOUT THE MOST COMMON SPECIES. IN GENERAL, WE THINK TABLE 1 SPEAKS FOR ITSELF, HOWEVER. IN ONLY TWO OF THE CYCLOPES EVENTS COULD CROSSING BE VREIFIED, AND TABLE 1 INCLUDES ONLY CROSSING EVENTS.

L140: Ok, so Table 1 only includes ‘confirmed crossings’ then? This needs to be made clearer.

WE ADDED CLARIFICATION IN THE TEXT AND THE TABLE HEADING.

L145: You call this a liana bridge here but I suggest that Artificial Canopy Bridge would be more consistent with your previous description.

THROUGHOUT THE TEXT WE NOW REFER TO THE LIANA BRIDGE AS A SEMI-ARTIFICIAL CANOPY BRIDGE.

L147: Suggest using common name, followed by scientific name.

FIXED

L154: If this section is to be included then it needs to be much more detailed. There is no breakdown of different costs and no financial figures presented at all. The proportional cost given is completely meaningless without knowing the total cost of the Sagari Project, which is undescribed and undefined.

THIS SECTION WAS CUT FOR LACK OF MORE DETAILED DATA.

L159: The meaning of this sentence is not at all clear.

WE DELETED THIS SENTENCE ALONG WITH THE REST OF THE PARAGRAPH.

Discussion

It would be useful to widen the discussion to consider techniques employed in other regions of the world, including other habitats, to improve connectivity in fragmented landscapes e.g. corridors, tunnels, fence gaps etc.

WE ADDED A SENTENCE ABOUT OTHER TECHNIQUES BUT DIDN’T FEEL WE HAD THE SPACE TO GO INTO GREAT DETAIL.

L178: This could have been tested (at least minimally) by also presenting an alternative ACB made with man-made materials. It is important to recognise that the natural material of a dead liana had a limited lifetime. So even, if having a quicker uptake than a bridge made of artificial materials, what is the long-term potential for an artificial bridge made of natural materials in any practical sense? (I see this point is recognised to some degree in L192 but the authors still maintain that the liana bridge is a viable option considering its lower (although unquantified) cost. The cost of constantly replacing a liana bridge would have to be considered here to give this comparison some more validity.

WE AGREE THAT A MORE SUBSTATIAL TEST WOULD BE VALUABLE. THIS WAS A PILOT STUDY. WE PLAN TO DO MORE TESTING AS THE OPPORTUNITY ARISES. WE ADDED MORE INFORMATION TO THE DISCUSSION OF THIS POINT.

L180: Which primates?

DETAIL ADDED.

L200-209: This situation is important to discuss in general but of less relevance to the more theoretical questions about connectivity. The temporary fragmentation caused by the RoW could perhaps be better seen as an opportunity to test how bridges could reduce the effects of fragmentation from road construction.

WE EDITED THIS PARAGRAPH TO MAKE THE CONCLUSIONS MORE RELEVANT TO A BROADER AUDIENCE.

Figures

Figure 1: Very nice photos! Also useful to appreciate the methodology.

THANK YOU!

Figure 2/3: These are nice photos too but may be more appropriate as supplementary figures. It may be more useful to instead include figures that present the main findings of the study.

WE MOVED THESE FIGURES TO THE SUPPLEMENTARY INFORMATION AND ADDED TWO FIGURES TO ILLUSTRATE THE METHODS AND RESULTS.

Tables

Table 1: Need to specify in the table title that this only includes results from the Natural Canopy Bridges. Where is Pagoreni? If comparisons to this site comprise an important aspect of this study, then the details should also be provided in the Methods. Perhaps this site (if nearby) could be included in the suggested map figure.

BASED ON COMMENTS FROM REVIEWER 1 AND 3, WE DECIDED TO RESTRICT THIS COMPARISON TO THE DISCUSSION.

What was the sampling effort (number of bridges, number of camera traps, length of study period etc.) at this site?

THERE WERE 13 BRIDGES MONITORED FOR A YEAR AT PAGORENI. AGAIN, THE COMPARISON WAS MOVED TO THE DISCUSSION.

Do the rates represent events or numbers of individuals? I.e. did the group of night monkeys score as 1 event or did each individual score separately? 

THEY REPRESENT GROUPS

Suggest a more meaningful order for the species e.g. arranged taxonomically.

THE ORDER IS BY NUMBER OF CROSSING EVENTS, ALLOWING FOR THE READERS’ ATTENTION TO BE FOCUSED ON WHICH SPECIES USED THE BRIDGES MOST.

Do the species names need associated common names and/or authorities?

COMMON NAMES ADDED

Table 2: This table for ACB presents data on crossing/not-crossing that seems to be missing for the NCBs. However, in the opposite manner, this table does not distinguish between diurnal and nocturnal crossings. Suggest greater attention to present comprehensive and consistent results.

TABLES 1 AND 2 NOW SHOW THE SAME INFORMATION FOR THE NCBS AND SEMI-ACB, RESPECTIVELY. HOWEVER, FOR TABLE 2 WE DID NO DISTINGUISH BETWEEN DAY AND NIGHT BECAUSE ALL BUT ONE EVENT HAPPENED AT NIGHT (SEE TABLE HEADING).

Supplementary Materials These were not accessible to the reviewer.

THERE WERE NONE IN THE PREVIOUS DRAFT

Funding If funded by the construction company involved, does this not represent a potential conflict of interest?"

WE ARE CONFIDENT THAT THERE IS NOT A CONFLICT OF INTEREST IN OUR WORK. EVEN THOUGH THE CONSTRUCTION COMPANY FUNDED THE STUDY, ALL ASPECTS OF THE DESIGN, METHODS, AND DATA INTERPRETATION WERE CONDUCTED BY SMITHSONIAN AND WALSH STAFF. COMPANY PERSONNEL WERE INVOLVED IN MANAGING LOGISTICS AND PERSONAL SAFETY DURING FIELD WORK. A REPSOL STAFF MEMBER ALSO CONTRIBUTED TO THE DEVELOPMENT OF THIS MANUSCRIPT.

Reviewer 2 Report

Introduction 

Authors examine (14 months) natural and artificial canopy bride use by arboreal mammals to connect fragmented habitats resulting from linear infrastructure. 

Overall: natural canopy bridges (NCBs) 55 crossings/100 trap nights by 16 species; and artificial canopy bridges (ACBs) 90 crossings/100 trap nights by five species.

Objectives: 

1) NCBs are used frequently and by a broad range of arboreal mammal species

2) Test use of an ACB made of liana by arboreal mammals

3) Provide general information on project costs

Methods

Line 94: delete “It.”

Discussion

NCBs were used frequently and by a broad range of arboreal mammal species

ACBs used quickly after installation and may be a good option if NCBs aren’t a possibility

Cb selection should be based on project duration and forest succession and pioneer species may connect forest

Author Response

AUTHOR RESPONSES IN CAPS

REVIEWER 2

               Yes         Can be improved              Must be improved           Not applicable

Does the introduction provide sufficient background and include all relevant references?              

(x)           ( )           ( )           ( )

Is the research design appropriate?                         (x)           ( )           ( )           ( )

Are the methods adequately described?                (x)           ( )           ( )           ( )

Are the results clearly presented?                            (x)           ( )           ( )           ( )

Are the conclusions supported by the results?     (x)           ( )           ( )           ( )

Comments and Suggestions for Authors

Introduction

Authors examine (14 months) natural and artificial canopy bride use by arboreal mammals to connect fragmented habitats resulting from linear infrastructure.

Overall: natural canopy bridges (NCBs) 55 crossings/100 trap nights by 16 species; and artificial canopy bridges (ACBs) 90 crossings/100 trap nights by five species.

THANK YOU FOR YOUR REVIEW OF OUR MANUSCRIPT.

Objectives:

1) NCBs are used frequently and by a broad range of arboreal mammal species

2) Test use of an ACB made of liana by arboreal mammals

3) Provide general information on project costs

Methods

Line 94: delete “It.”

FIXED

Discussion

NCBs were used frequently and by a broad range of arboreal mammal species

ACBs used quickly after installation and may be a good option if NCBs aren’t a possibility

Cb selection should be based on project duration and forest succession and pioneer species may connect forest

Reviewer 3 Report

This is a really interesting piece of work and a valuable contribution to the literature. Natural canopy bridges provide a really elegant solution to fragmentation by linear infrastructure, particularly if existing animal movement pathways are identified and retained. The example of the artificial bridge made of liana is fantastic. It shows that mitigation solutions do not have to be expensive or over-engineered to be effective for wildlife. This is the kind of agile thinking and creative conservation action we need to see described and tested far more frequently in the scientific literature. 

Some aspects of the introduction and discussion present the work from a slightly localised point of view, with many statements and references quite specific to their area. I’ve made some suggestions of how this might be broadened out without putting a strain on the word count. This work has relevance beyond tropical rainforests and gas pipelines, but I think a little bit of revision is required to highlight this. 

One of the stated aims is to assess the project cost. However no information on cost is provided. The authors use very vague statements like ‘expensive’ or ‘relatively cheap’, both throughout the introduction and the results, but don’t really give people an indication of what that means. Cheap to one person may be expensive to another. For example, Weston 2011 is cited as an example of ‘extremely  expensive’ yet the crossing structures in that paper range from $1000 to $17,500. Relative to the cost of the entire construction project (often many $100,000s, even millions) or the cost of other measures such as land-bridges or underpasses (millions), a rope ladder isn’t extremely expensive. 

Despite it being an aim, the authors provide no information on the project cost, except that it is 0.12% of the total budget (a number that is not provided). I would have expected to see a breakdown of the costs of an individual natural canopy bridge (or the range), a liana bridge, and then the associated installation/monitoring/maintenance costs provided separately. If the authors cannot provide this information, I suggest they remove this as an aim and shift these comments to the discussion. 

The rest of my comments are minor and relate to making things clearer, expanding the references or highlighting the broader implications of the work .

Minor comments and clarifications

Introduction

Ln 30 - The first sentences of the introduction set up the problem of fragmentation due to linear infrastructure, but perhaps provide an overly simplified view. e.g. Habitat loss is the first impact felt here (the area cleared to accomdate the pipe) but this impact is not really acknowledged. Also, the authors refer to the RoW ‘splitting the forest in two’, which is perhaps specific to their study area - in most cases across the world, there are unfortunately many more subdivisions…Perhaps it’s more accurate here to simply say that the “RoW, like a road, divides the canopy”. 

The authors also talk about this fragmentation affects arboreal species but they do not explain how. Is it due to a behavioural avoidance of the open space? Or because the gap in suitable habitat is too wide to be crossed, and so movement is affected? I assume the latter, but to help the work be more relevant to a global audience it’s important to clarify - because only one of these problems can be addressed by a canopy bridge.

Ln 36 - “...have proven to be an effective way to mitigate this form of forest fragmentation and restore canopy connectivity in forested areas impact by linear infrastructure [e.g. 1-5].” Should reference Soanes et al 2018 here, as I think it is the only study that has successfully demonstrated restoration of canopy connectivity through artificial bridges using genetic data and before-after comparisons. 

SoanesK, Taylor AC, Sunnucks P, et al.2018. Evaluating the success of wildlife crossing structures using genetic approaches and an experimental design: Lessons from a gliding mammal Journal of Applied Ecology 55: 129–138.

Ln 39. The sentence starting with “They can be simple…” is a bit hard to read as it presents two lists within a list almost. A semicolon before “or extremely expensive” may help more clearly show this. 

Methods

Under 2.1, Study site - can you provide information on the overall width of the pipeline here?

2.2 Bridge selection - did wildlife monitoring or existing movement paths influence site selection or were the criteria primarily based on engineering feasibility? Perhaps you could highlight in the discussion how the effectiveness of these bridges could be even greater if ecological factors play a larger role in the selection of sites. 

2.3 “Natural liana bridge” is the subheading, but you refer to this as an artificial bridge elsewhere. 

I’m curious as to why this site was selected - was it a desired site for natural bridge connection but the branches has broken? was it subject to the same site selection criteria as natural canopy bridges? 

2.4 Canopy bridge monitoring

It’s unclear to me how the cameras were set, or how there are 1-3 possible crossing locations per ‘structure’. I assume that this has something to do with how natural canopy bridges are created, possibly consisting of multiple branches? A diagram or close up image of exactly what these structures look like and how the cameras were placed to capture animals crossing would be really valuable to help readers picture the study. This is a fairly unique mitigation method, and many will not be familiar with exactly how they look. It could be supplementary material if space is an issue. 

There should be more information on how crossings were verified. What were the criteria? Was it based on two-camera confirmation, the animals behaviour and direction of travel, or a combination? Or was it simply all the events that weren’t clear ‘turn backs’/explorations? 

Results

3.1 Canopy bridge monitoring

I’m curious about the reliability of your cameras. Is the total number of trap nights provided based on actual known nights where the cameras were operational, or were cameras simply assumed to be operational the entire time e.g. were they functioning and SD cards still had room left to record when checked in March 2018? If not, it may indicate that the monitoring effort was less than expected (i.e. if a memory card filled up days or months prior to the camera being checked, then it is no longer operational). 

Ln 127 “There was a great difference…” The language here can be a bit ambiguous. “great” can also mean “excellent” or “good”. Stick to clearer words like “large” in the results. 

I would like to see the number of crossings per structure provided either in the main results or as supplementary material. This would help support the authors statements about variability between structures, and ensure that the data is more easily comparable to other studies (or included in a meta-analysis). 

It is more common to present crossing rates per day/night, rather than per 100 nights. But if the authors are concerned that this would make the numbers too low with too many decimals, then per 100 nights is probably ok. 

3.3 Project costs

See comments above - if this is to be a section of the results, and an aim, I should think the actual project costs should be presented here. Readers would expect to learn roughly how much it costs to retain a natural canopy bridge and how much it costs to build an artificial/liana bridge. Perhaps also some information on the maintenance and other costs (separate from materials/construction). If this is not possible for commercial in-confidence reasons, then I don’t think it should be presented as an aim or result. 

Table 1. I don’t understand why data from the Pagoreni study (published elsewhere in 2017) is presented alongside the results from this study. This may be a meaningful comparison in a local context, but is not as relevant to a broader audience. I think the authors can simply refer to these findings in the discusison, rather than present them again here. Presenting them alongside one another implies that there is an important difference between the two sites that might drive differences in use - it invites the readers to make comparisons and look for explanations. But there isn’t really enough data to explain why these differences may occur (local species abundance, placement of structures within movement paths, individual variability, structural or environmental attributes etc) so such a comparison is not really appropriate or informative at this point. 

If there is another reason for presenting the data here, could the authors explain this more or highlight it’s relevance?

Discussion

Related to the point regarding table 1, I’m not convinced that the authors have enough information to try and explain why the 2017 study of a different area (Pagoreni) and the current study yeilded slightly different results. I’m also not sure that this is particularly interesting outside of a very specific local context. The differences in usage rates could be due to a large number of factors that were not measured (or at least not presented) by the authors - species diversity at each site, local abundance/population size, position of bridges relative to home ranges or movement paths, design of the bridges, environmental conditions surrounding the bridges, time of year, even inter-individual variations in animal behaviour etc. Given that we know that differences even between bridges in the same area are large, it seems odd to try and present and explain the difference between two different areas/forests without having at least some of this additional information. 

I think the authors could instead focus on the results within their own study, and can note that they are generally comparable to to other studies on arboreal mammals using crossing structures (e.g. similar species sets observed in Gregory et al 2017, high variation between sites has been observed by many studies including Goldingay et al 2013 and Soanes et al 2015)

Ln 176 - In reference to habituation times, Soanes et al 2013 explicitly monitored and demonstrated the change in use over time to identify a habituation period, while Valladares-Padua 1995 describe the pole bridge being used ‘as soon as it was assembled’

Paragraph starting Ln 182 - I think the discussion of how forest type can influence the feasibility of using natural canopy bridges is an excellent point to make. Again though, it is unclear why this is set up as a comparison to the 2017 study. The authors could make the point more generally about how forest type is an important factor to consider when deciding whether or not natural canopy bridges are a viable mitigation option - this would be of interest to a broader range of managers and researchers. 

Otherwise this is a really nice discussion on feasibility. 

Ln 188 - “For this reason, studies evaluating the use of different substrates and designs are needed”. I don’t think “substrates” is the right word here. Can you use “materials” instead? Substrates makes me think of the ground, or a space that things grow on. 

Ln 208 - The authors seem to imply here that natural canopy bridges aren’t as necessary when the forest is going to regenerate over the cleared area in time and ‘reconnect’ itself. Time lag could be something to clarify here. In many parts of the world, an impact on tree canopy would take several decades, potentially closer to 100 years to restore on it’s own. This would leave the site impacting local wildlife for several generations, and potentially be enough to cause lasting damage. Could the authors add some discussion or caveats here to provide guidance to a broader audience than just tropical forests?

Ln 212 - Appropriate distance is a key issue, and one discussed at length in Bissonette and Adair 2008. You may be able to draw on this work to provide more weight to your arguments here. 

Bissonette JA and Adair W. 2008. Restoring habitat permeability to roaded landscapes with isometrically-scaled wildlife crossings Biol Conserv 141: 482-488.

Author Response

AUTHOR RESPONSES IN CAPS

REVIEWER 3

               Yes         Can be improved              Must be improved           Not applicable

Does the introduction provide sufficient background and include all relevant references?              

( )           (x)           ( )           ( )

Is the research design appropriate?                         (x)           ( )           ( )           ( )

Are the methods adequately described?                ( )           (x)           ( )           ( )

Are the results clearly presented?                            (x)           ( )           ( )           ( )

Are the conclusions supported by the results?     ( )           (x)           ( )           ( )

Comments and Suggestions for Authors

This is a really interesting piece of work and a valuable contribution to the literature. Natural canopy bridges provide a really elegant solution to fragmentation by linear infrastructure, particularly if existing animal movement pathways are identified and retained. The example of the artificial bridge made of liana is fantastic. It shows that mitigation solutions do not have to be expensive or over-engineered to be effective for wildlife. This is the kind of agile thinking and creative conservation action we need to see described and tested far more frequently in the scientific literature.

THANK YOU FOR RECOGNIZING THE VALUE OF THE STUDY. WE APPRECIATE THE POSITIVE FEEDBACK.

Some aspects of the introduction and discussion present the work from a slightly localised point of view, with many statements and references quite specific to their area. I’ve made some suggestions of how this might be broadened out without putting a strain on the word count. This work has relevance beyond tropical rainforests and gas pipelines, but I think a little bit of revision is required to highlight this.

THANK YOU FOR THE FEEDBACK. WE HAVE FOLLOWED YOUR SUGGESTIONS THROUGHOUT THE MANUSCRIPT.

One of the stated aims is to assess the project cost. However no information on cost is provided. The authors use very vague statements like ‘expensive’ or ‘relatively cheap’, both throughout the introduction and the results, but don’t really give people an indication of what that means. Cheap to one person may be expensive to another. For example, Weston 2011 is cited as an example of ‘extremely  expensive’ yet the crossing structures in that paper range from $1000 to $17,500. Relative to the cost of the entire construction project (often many $100,000s, even millions) or the cost of other measures such as land-bridges or underpasses (millions), a rope ladder isn’t extremely expensive.

Despite it being an aim, the authors provide no information on the project cost, except that it is 0.12% of the total budget (a number that is not provided). I would have expected to see a breakdown of the costs of an individual natural canopy bridge (or the range), a liana bridge, and then the associated installation/monitoring/maintenance costs provided separately. If the authors cannot provide this information, I suggest they remove this as an aim and shift these comments to the discussion.

THANK YOU FOR THIS CONSTRUCTIVE FEEDBACK. WE HAVE DELETED THIS AIM AND THE SECTION IN THE RESULTS.

The rest of my comments are minor and relate to making things clearer, expanding the references or highlighting the broader implications of the work.

Minor comments and clarifications

Introduction

Ln 30 - The first sentences of the introduction set up the problem of fragmentation due to linear infrastructure, but perhaps provide an overly simplified view. e.g. Habitat loss is the first impact felt here (the area cleared to accomdate the pipe) but this impact is not really acknowledged. Also, the authors refer to the RoW ‘splitting the forest in two’, which is perhaps specific to their study area - in most cases across the world, there are unfortunately many more subdivisions…Perhaps it’s more accurate here to simply say that the “RoW, like a road, divides the canopy”.

WE ADDED AN INTRODUCTORY PARAGRAPH ABOUT THE THREATS OF HUMAN DEVELOPMENT. WE ALSO CHANGED THE WORDING TO REFLECT YOUR SUGGESTION.

The authors also talk about this fragmentation affects arboreal species but they do not explain how. Is it due to a behavioural avoidance of the open space? Or because the gap in suitable habitat is too wide to be crossed, and so movement is affected? I assume the latter, but to help the work be more relevant to a global audience it’s important to clarify - because only one of these problems can be addressed by a canopy bridge.

WE ADDED SOME DETAIL HERE TO ADDRESS THIS QUESTION. THE REVERENCE CITED [GREGORY ET AL 2017] ADDRESSES THE POINT IN MORE DETAIL SO WE DID NOT FEEL IT NECESSARY TO LIST EXAMPLES, ALSO DUE TO LIMITED SPACE.

Ln 36 - “...have proven to be an effective way to mitigate this form of forest fragmentation and restore canopy connectivity in forested areas impact by linear infrastructure [e.g. 1-5].” Should reference Soanes et al 2018 here, as I think it is the only study that has successfully demonstrated restoration of canopy connectivity through artificial bridges using genetic data and before-after comparisons.

SoanesK, Taylor AC, Sunnucks P, et al.2018. Evaluating the success of wildlife crossing structures using genetic approaches and an experimental design: Lessons from a gliding mammal Journal of Applied Ecology 55: 129–138.

REFERNENCE ADDED. THANK YOU.

Ln 39. The sentence starting with “They can be simple…” is a bit hard to read as it presents two lists within a list almost. A semicolon before “or extremely expensive” may help more clearly show this.

WE BROKE THIS SENTENCE INTO TWO TO MAKE IT CLEARER

Methods

Under 2.1, Study site - can you provide information on the overall width of the pipeline here?

ADDED.

2.2 Bridge selection - did wildlife monitoring or existing movement paths influence site selection or were the criteria primarily based on engineering feasibility? Perhaps you could highlight in the discussion how the effectiveness of these bridges could be even greater if ecological factors play a larger role in the selection of sites.

WHILE THIS TYPE OF EVALUATION WOULD HAVE BEEN EXTREMELY VALUABLE (AND WE CONSIDERED IT EXTENSIVELY) IT WAS NOT FEASIBLE. WE IMAGINE THAT FOR SUCH AN EVALUATION TO BE COMPREHENSIVE, IT WOULD HAVE TO BE EXTREMELY BROAD IN SCALE, GIVEN THE LENGTH OF THE PIPELINE PATH. WE ADDED MORE INFORMATION ON THIS POINT IN THE METHODS AND ALSO ADDRESS IT IN THE DISCUSSION AS A SUGGESTION FOR FUTURE STUDIES.

2.3 “Natural liana bridge” is the subheading, but you refer to this as an artificial bridge elsewhere.

I’m curious as to why this site was selected - was it a desired site for natural bridge connection but the branches has broken? was it subject to the same site selection criteria as natural canopy bridges?

WE HAVE CHANGED THE SUBHEADING AND ALL REFERENCES TO THE LIANA BRIDGE TO “SEMI-ARTIFICIAL CANOPY BRIDGE” TO REDUCE CONFUSION. INSTALLING THE LIANA BRIDGE WAS A PILOT EXPERIMENT GIVEN EXACTLY THE RIGHT CIRCUMSTANCES. WE HAD NOT PREVIOUSLY PLANNED TO DO THE INSTALLATION, BUT FINDING THE RIGHT CONDITIONS—A LIANA IN GOOD CONDITION, CUT AND HANGING RIGHT AT THE SIDE OF THE ROW—WE DECIDED TO TEST AN IDEA WE HAD HELD FOR A LONG TIME. AT THIS SITE, THERE ARE LIANAS LINING THE ROW IN MANY PLACES, SO THE CONTEXT ALLOWED FOR THE EXPERIMENT, BUT WE DID NOT HAVE THE RESOURCES OR TIME TO INSTALL MORE THAN ONE. WE ADDED A PHRASE (“WE FOUND THE OPTIMAL CONDITIONS IN WHICH TO CREATE A SEMI-ACB…”) TO ADDRESS THIS POINT.

2.4 Canopy bridge monitoring

It’s unclear to me how the cameras were set, or how there are 1-3 possible crossing locations per ‘structure’. I assume that this has something to do with how natural canopy bridges are created, possibly consisting of multiple branches? A diagram or close up image of exactly what these structures look like and how the cameras were placed to capture animals crossing would be really valuable to help readers picture the study. This is a fairly unique mitigation method, and many will not be familiar with exactly how they look. It could be supplementary material if space is an issue.

WE ADDED DETAIL TO THE TEXT. YES, THE BRANCHES OFTEN CONNECT AT MORE THAN ONE POINT. WE ALSO ADDED A FIGURE TO HELP ILLUSTRATE WHAT THE BRIDGES LOOK LIKE AND HOW CAMERAS WERE PLACED.

There should be more information on how crossings were verified. What were the criteria? Was it based on two-camera confirmation, the animals behaviour and direction of travel, or a combination? Or was it simply all the events that weren’t clear ‘turn backs’/explorations?

WE ADDED DETAIL TO THE TEXT. WE USED JUST ONE CAMERA PER CROSSING POINT. WE COUNTED A CROSSING EVENT WHEN IT WAS CLEAR THAT THE ANIMAL DID NOT DOUBLE BACK.

Results

3.1 Canopy bridge monitoring

I’m curious about the reliability of your cameras. Is the total number of trap nights provided based on actual known nights where the cameras were operational, or were cameras simply assumed to be operational the entire time e.g. were they functioning and SD cards still had room left to record when checked in March 2018? If not, it may indicate that the monitoring effort was less than expected (i.e. if a memory card filled up days or months prior to the camera being checked, then it is no longer operational).

WE PROGRAM THE CAMERAS TO TAKE ONE PHOTO DAILY TO VERIFY THAT THEY ARE FUNCTIONING (RECONYX CAMERAS HAVE VERY HIGH QUALITY PARTS AND IN OUR EXPERIENCE ONLY FAIL UNDER EXTREME CIRCUMSTANCES; WHEN THIS HAPPENS, IT IS OBVIOUS BECAUSE THEY STOP TAKING PICTURES). INDEED ON TWO OCCASIONS A CAMERA’S CARD FILLED. WE ADDED MORE INFORMATION ON THIS POINT.

Ln 127 “There was a great difference…” The language here can be a bit ambiguous. “great” can also mean “excellent” or “good”. Stick to clearer words like “large” in the results.

FIXED

I would like to see the number of crossings per structure provided either in the main results or as supplementary material. This would help support the authors statements about variability between structures, and ensure that the data is more easily comparable to other studies (or included in a meta-analysis).

WE ASSUME YOU MEAN PER BRIDGE? WE HAVE ADDED A FIGURE SHOWING CROSSING RATES AND NUMBERS OF SPECIES PER BRIDGE TO ILLUSTRATE THIS.

It is more common to present crossing rates per day/night, rather than per 100 nights. But if the authors are concerned that this would make the numbers too low with too many decimals, then per 100 nights is probably ok.

WE HAVE SEEN BOTH BUT CAMERA TRAPPING TEXTS (E.G. ROVERO AND ZIMMERMAN 2016 Camera Trapping for Wildlife Research) OFTEN SUGGEST RATES PER 100 DAYS/NIGHTS (OFTEN CALLED THE RELATIVE ABUNDANCE INDEX OR RAI) PRESUMABLY BECAUSE IT MAKES THE NUMBERS EASIER TO CONCEPTUALIZE FOR A READER.

3.3 Project costs

See comments above - if this is to be a section of the results, and an aim, I should think the actual project costs should be presented here. Readers would expect to learn roughly how much it costs to retain a natural canopy bridge and how much it costs to build an artificial/liana bridge. Perhaps also some information on the maintenance and other costs (separate from materials/construction). If this is not possible for commercial in-confidence reasons, then I don’t think it should be presented as an aim or result.

AS RECOMMENDED, WE REMOVED THIS TOPIC FROM THE OBJECTIVES AND RESULTS.

Table 1. I don’t understand why data from the Pagoreni study (published elsewhere in 2017) is presented alongside the results from this study. This may be a meaningful comparison in a local context, but is not as relevant to a broader audience. I think the authors can simply refer to these findings in the discusison, rather than present them again here. Presenting them alongside one another implies that there is an important difference between the two sites that might drive differences in use - it invites the readers to make comparisons and look for explanations. But there isn’t really enough data to explain why these differences may occur (local species abundance, placement of structures within movement paths, individual variability, structural or environmental attributes etc) so such a comparison is not really appropriate or informative at this point.

If there is another reason for presenting the data here, could the authors explain this more or highlight it’s relevance?

WE MADE THIS COMPARISON BECAUSE THE STUDY SITES ARE VERY SIMILAR (IN FACT ALL PART OF THE SAME PIPELINE PROJECT. BUT WE UNDERSTAND THE CONFUSION AND THIS COMPARISON WAS REMOVED FROM THE RESULTS AND INCLUDED ONLY IN THE DISCUSSION.

Discussion

Related to the point regarding table 1, I’m not convinced that the authors have enough information to try and explain why the 2017 study of a different area (Pagoreni) and the current study yeilded slightly different results. I’m also not sure that this is particularly interesting outside of a very specific local context. The differences in usage rates could be due to a large number of factors that were not measured (or at least not presented) by the authors - species diversity at each site, local abundance/population size, position of bridges relative to home ranges or movement paths, design of the bridges, environmental conditions surrounding the bridges, time of year, even inter-individual variations in animal behaviour etc. Given that we know that differences even between bridges in the same area are large, it seems odd to try and present and explain the difference between two different areas/forests without having at least some of this additional information.

I think the authors could instead focus on the results within their own study, and can note that they are generally comparable to to other studies on arboreal mammals using crossing structures (e.g. similar species sets observed in Gregory et al 2017, high variation between sites has been observed by many studies including Goldingay et al 2013 and Soanes et al 2015)

WE HAVE CUT DOWN THE COMPARISON TO PAGORENI AND ADDED A BIT MORE EVALUATION OF THIS STUDY, AS SUGGESTED.

Ln 176 - In reference to habituation times, Soanes et al 2013 explicitly monitored and demonstrated the change in use over time to identify a habituation period, while Valladares-Padua 1995 describe the pole bridge being used ‘as soon as it was assembled’

CITATIONS ADDED

Paragraph starting Ln 182 - I think the discussion of how forest type can influence the feasibility of using natural canopy bridges is an excellent point to make. Again though, it is unclear why this is set up as a comparison to the 2017 study. The authors could make the point more generally about how forest type is an important factor to consider when deciding whether or not natural canopy bridges are a viable mitigation option - this would be of interest to a broader range of managers and researchers.

Otherwise this is a really nice discussion on feasibility.

WE EDITED THIS PARAGRAPH TO ADDRESS THIS POINT.

Ln 188 - “For this reason, studies evaluating the use of different substrates and designs are needed”. I don’t think “substrates” is the right word here. Can you use “materials” instead? Substrates makes me think of the ground, or a space that things grow on.

CHANGED

Ln 208 - The authors seem to imply here that natural canopy bridges aren’t as necessary when the forest is going to regenerate over the cleared area in time and ‘reconnect’ itself. Time lag could be something to clarify here. In many parts of the world, an impact on tree canopy would take several decades, potentially closer to 100 years to restore on it’s own. This would leave the site impacting local wildlife for several generations, and potentially be enough to cause lasting damage. Could the authors add some discussion or caveats here to provide guidance to a broader audience than just tropical forests?

WE EDITED THIS PARAGRAPH TO MAKE THE CONCLUSIONS MORE RELEVANT TO A BROADER AUDIENCE.

Ln 212 - Appropriate distance is a key issue, and one discussed at length in Bissonette and Adair 2008. You may be able to draw on this work to provide more weight to your arguments here.

Bissonette JA and Adair W. 2008. Restoring habitat permeability to roaded landscapes with isometrically-scaled wildlife crossings Biol Conserv 141: 482-488.

WE HAVE ADDED A SENTENCE ABOUT THIS STUDY.

Round 2

Reviewer 1 Report

The revised manuscript is much improved, having taken on board most of the modifications suggested by my review and the other reviewers. However, the results still lack any serious attempt to include statistical analyses to test any hypotheses about variation in crossing events/rates between species or bridge types. There is a mention of low sample sizes in relation to the question of inter-bridge distance but perhaps this needs to be generalised to help explain the absence of any hypothesis-driven statistical analyses, and to justify the focus instead on a more descriptive summary.

There also remain a few more minor points to address:

Introduction – the definition of ‘Right of Way’ that you provide in your response is useful and I think including a summary of this in the introduction could be helpful for the general reader. Perhaps these details could be included in the Supplementary Materials, together with the timeline of the pipeline construction and the lifespan of the different bridges.

Methods – the inclusion of camera failures/batteries/memory cards is helpful. Are these issues fully taken account of when presenting the number of trap day/nights?

Results – perhaps it would be of interest (at least in the supplementary materials?) to present the results for other taxonomic groups i.e. birds and reptiles.

Table 1 – I still maintain that a taxonomic order would be much more useful for the reader than the current order; the table title still needs to specify that these results are for NCBs.

Table 2 – Again, a taxonomic order would be more useful here. Suggest “ one of the two events for Sapajus macrocephalus”.

Figure 1 – specify symbols for the two sites in the inset.

Figure 4 – suggest separate plots i.e. a two-panel plot – using multiple axes on same plot is not generally considered to be a clear way of presenting results.

Conflict of interest – I suggest that it would be useful to include a statement along the lines of that in your response to clarify the situation.

Author Response

Reviewer 1

Comments and Suggestions for Authors

The revised manuscript is much improved, having taken on board most of the modifications suggested by my review and the other reviewers. However, the results still lack any serious attempt to include statistical analyses to test any hypotheses about variation in crossing events/rates between species or bridge types. There is a mention of low sample sizes in relation to the question of inter-bridge distance but perhaps this needs to be generalised to help explain the absence of any hypothesis-driven statistical analyses, and to justify the focus instead on a more descriptive summary.

THANK YOU FOR THIS REVIEW AND THE FEEDBACK ON DATA ANALYSES. THE OBJECTIVE OF THIS STUDY WAS TO VERIFY THAT NCBS AND SACBS ARE USED BY ARBOREAL MAMMALS. WE DID NOT PRESENT HYPOTHESES ON DIFFERENCES IN USE AMONG BRIDGES, BRIDGE TYPES, OR SPECIES BECAUSE THIS WAS NOT THE OBJECTIVE OF THIS STUDY. HAD THE FOREST LENT ITSELF TO THE PRESENCE OF MORE NCBS, HAD THE SITUATION LENT ITSELF TO A BEFORE, DURING, AFTER STUDY, OR HAD THE BUDGET ALLOWED INSTALLATION OF MORE LIANA BRIDGES, PERHAPS WE WOULD HAVE BEEN ABLE TO FORM HYPOTHESES. BUT AGAIN, GIVEN VERY LOW SAMPLE SIZES OF THE TWO BRIDGE TYPES (7 AND 1) AND THE LIKELY STRONG EFFECTS OF UNMEASURABLE VARIABLES ON INTERSPECIFIC DIFFERENCES IN USE (E.G. THE LOCAL DENSITY OF EACH SPECIES, WHETHER A PATH FORMED BY THE BRANCHES IS PART OF A PREEXISTING PATH FOR ONE GROUP OR THE OTHER OF A SPECIES, ETC), WE DO NOT THINK THE DATA LEND THEMSELVES TO IN-DEPTH HYPOTHESIS TESTING OR STATISTICAL ANALYSES AND FEAR THAT ADDING ANALYSES WOULD BE AN OVERSTATEMENT OF THE POWER OF THE DATA. WE ADDED AN ADDITIONAL SENTENCE TO THE RESULTS SECTION TO ADDRESS THIS CONCERN.

There also remain a few more minor points to address:

Introduction – the definition of ‘Right of Way’ that you provide in your response is useful and I think including a summary of this in the introduction could be helpful for the general reader. Perhaps these details could be included in the Supplementary Materials, together with the timeline of the pipeline construction and the lifespan of the different bridges.

WE ADDED A BRIEF DESCRIPTION OF WHAT THE ROW IS IN THE SECOND PARAGRAPH OF THE INTRODUCTION. UNFORTUNATELY, WE DO NOT HAVE ACCESS TO A DETAILED TIMELINE OF PIPELINE CONSTRUCTION (THERE ARE DOZENS OF STEPS) IN THE FIVE DAYS WE HAVE TO RESUBMIT THE MANUSCRIPT.

Methods – the inclusion of camera failures/batteries/memory cards is helpful. Are these issues fully taken account of when presenting the number of trap day/nights?

YES, THE TOTAL NUMBER OF TRAP NIGHTS EXCLUDES NIGHTS WHEN A CAMERA WAS NOT FUNCTIONING OR A BRIDGE HAD BROKEN. WE ADDED A BRIEF COMMENT ON THIS TO THE RESULTS.

Results – perhaps it would be of interest (at least in the supplementary materials?) to present the results for other taxonomic groups i.e. birds and reptiles.

WE ADDED A SUPPLEMENTARY TABLE WITH INFO ON THE BIRDS AND REPTILES.

Table 1 – I still maintain that a taxonomic order would be much more useful for the reader than the current order; the table title still needs to specify that these results are for NCBs.

WE HAVE MADE THE SUGGESTED CHANGES, ALTHOUGH WE THINK THE TABLE IN ITS CURRENT STATE GIVES DISPROPORTIONATE EMPHASIS TO SPECIES LIKE CEBUS ALBIFRONS, WHICH WAS INVOLVED IN ONLY ONE EVENT.

Table 2 – Again, a taxonomic order would be more useful here. Suggest “ one of the two events for Sapajus macrocephalus”.

CHANGES MADE.

Figure 1 – specify symbols for the two sites in the inset.

MORE DETAIL ADDED.

Figure 4 – suggest separate plots i.e. a two-panel plot – using multiple axes on same plot is not generally considered to be a clear way of presenting results.

CHANGES MADE.

Conflict of interest – I suggest that it would be useful to include a statement along the lines of that in your response to clarify the situation.

STATEMENT ADDED.
